# The 1000+ mouse project for large-scale spatiotemporal parametrization and modeling of preclinical cancer immunotherapies

Adam L Kenet[1†], Sooraj Achar[1,2†], Alka Dwivedi[3], John Buckley[3], Marie Pouzolles[3], Haying Qin[3‡], Christopher Chien[3], Naomi Taylor[3,4]*, Grégoire Y Altan-Bonnet[1]*

[1]Immunodynamics Group, Laboratory of Integrative Cancer Immunology, Center for Cancer Research, National Cancer Institute, Bethesda, United States; [2]Kennedy Institute of Rheumatology, Nuffield Department of Orthopaedics, Rheumatology and Musculoskeletal Sciences, University of Oxford, Oxford, United Kingdom; [3]Pediatric Oncology Branch, Center for Cancer Research, National Cancer Institute, Bethesda, United States; [4]Université de Montpellier, Institut de Génétique Moléculaire de Montpellier, Montpellier, France

**\*For correspondence:**
taylorn4@mail.nih.gov (NT);
gregoire.altan-bonnet@nih.gov
(GYA-B)

†These authors contributed equally to this work

**Present address:** ‡Cellular Therapy, R&D, Astrazeneca, Gaithersburg, United States

**Competing interest:** The authors declare that no competing interests exist.

## eLife Assessment

The authors developed a **fundamental** computational method, which is intended to automatically process bioluminescence imaging-derived tumor images across anatomical regions and over time. This allows quantitative analysis of such data, and the authors applied it to describe the spatiotemporal distribution of tumour cells in response to CD19-targeted CAR-T cells that contained either CD28 or 4-1BB costimulatory domains. Some operational limitations were identified, which relate to the pipeline's reliance on predefined regions of interest instead of aligning signal sites with anatomical information, scaling, and limitations in taking animal pose into account. Overall, the authors provide **compelling** evidence for the functionality of their computational approach towards automated analysis of bioluminescence imaging data, while applying it to a current topic of wide interest in cell therapy research.

**Abstract** Preclinical studies of chimeric antigen receptor (CAR)-T cell immunotherapies are often based on monitoring bioluminescent tumors implanted in mice to assess anti-tumor cytotoxicity. Here, we introduce maRQup (**m**urine **a**utomatic **R**adiance **Q**uantification and **p**arametrization), an easy-to-use method that automatically processes bioluminescent tumor images for quantitative analysis. We demonstrate the ability of maRQup to analyze CAR-T cell treatments over >1000 tumor-bearing mice. We compare CD19-targeting CAR-T cells comprising either a CD28 or a 4-1BB costimulatory domain, and found the former controlled the tumor burden better initially, while the latter reduced the frequency of tumor relapse. We also applied maRQup to demonstrate faster tumor growth during the initial growth phase as compared to the relapse phase and to spatiotemporally analyze the high variability in immunotherapeutic control of tumors, based on their anatomical location. maRQup provides quantitative and statistically-robust insights on preclinical experiments that will contribute to the optimization of immunotherapies.

## Introduction

Over the past decade, chimeric antigen receptor (CAR)-T cell immunotherapy has emerged as a highly effective treatment for hematological malignancies (*McClory and Maude, 2023*, *Cappell and Kochenderfer, 2023*). The field has seen the completion of numerous clinical trials—over 500 in the United States and nearly 1000 in Asia and Europe as of early 2025—exploring innovative constructs with therapeutic applications against cancer as well as in autoimmunity (*Kyverna Therapeutics, 2023*, *Juno Therapeutics, Inc, a Bristol-Myers Squibb Company, 2024*, *Müller et al., 2024*) and viral infections (e.g., HIV) (*Deeks, 2023*, *University of Pennsylvania, 2023*). However, CAR-T cell therapies still face shortcomings. CAR-T cell toxicity is often limiting, and a lack of persistence of functional CAR-T cells is associated with tumor relapse (*Cappell and Kochenderfer, 2023*). Given the rapid and diverse developments in this field, it is essential to be able to quantify and standardize the preclinical data for testing the design of CAR-T cell constructs and for engineering anti-tumor T cell and non-T cell immune subsets, thereby optimizing clinical trials.

The present standard for the preclinical in vivo evaluation of human CAR-T cells relies on the kinetic assessment of human tumor cells engineered to express the luciferase gene in immunodeficient mice, generally NOD.Cg-*Prkdc^{scid}Il2rg^{tm1Wjl}*/SzJ (NSG) mice (*Carceles-Cordon et al., 2016*). The growth and eradication of the tumor upon CAR-T cell adoptive transfer is then assessed by luminescence imaging (*Carceles-Cordon et al., 2016*, *Bhaumik and Gambhir, 2002*, *Weissleder and Ntziachristos, 2003*). This in vivo assay, which assesses tumor growth vs. tumor eradication and mouse survival, is generally required by the FDA for any investigational new drug (IND) application of a novel CAR-T cell design. However, it is concerning that there is significant variability in how these data are reported, highlighting the urgent need for better quantification, spatiotemporal resolution, and statistical analyses.

A limiting step for many experimentalists and clinicians is the lack of a framework to automatically process large in vivo experiments: visualization, quantitation, and analysis of these in vivo assays are often handled by manual copying and pasting of images followed by visual inspection. Here, we introduce a Python pipeline (https://github.com/soorajachar/radianceQuantifier copy archived at *Kenet and Achar, 2026*) named maRQup (for **m**urine **a**utomatic **R**adiance **Q**uantification and **p**arametrization) that automates the processing, spatiotemporal quantification, and modeling of in vivo bioluminescent tumor image data.

We applied this toolkit to assess the tumor dynamics of in vivo CAR-T cell-based experiments performed on >1000 NSG mice. This systematic endeavor afforded the opportunity to carry out advanced statistical analyses and mathematical modeling to spatiotemporally resolve the responsiveness of leukemia-bearing NSG mice to CAR-T cell immunotherapies. Using our quantitative framework, we have built a repository with over 7500 images, allowing for the large-scale and longitudinal evaluation of CAR-T cell therapy. Analysis and modeling of this database has revealed novel characteristics of CAR-T cell designs as well as novel dynamics of tumor responses that would not have been feasible from a manual evaluation of the data. Our quantitative method and repository will promote the identification of key differences in immune-based treatments in murine models, thereby leading to improved cancer immunotherapy trials for patients.

## Results

### maRQup pipeline

All IVIS images were processed, annotated, and analyzed using a custom-built Python pipeline named maRQup (*Figure 1*). Each IVIS image contains 1–5 mice, as mice in the same experimental group are typically imaged at the same time (*Figure 1a*). Thus, we developed an algorithm that automatically identifies and isolates each mouse from the group image to allow for further image processing and analysis (*Figure 1b*). In order to analyze tumor burden within different regions of each mouse (e.g., liver, lungs, bone marrow, and snout), all images were processed so that anatomical regions were aligned across mice: first, the tail of each mouse was automatically removed from each image as the positioning of the tail resulted in large variability in the length and width of each image; next, the images were scaled vertically to the same height to normalize mouse length and to ensure anatomical regions were horizontally aligned. Additionally, maRQup removes from analysis any mice that were slanted due to misplacement during image acquisition (<1.2% (91/7633) of the images in our experience). After image processing and alignment of anatomical regions, the average radiance per pixel

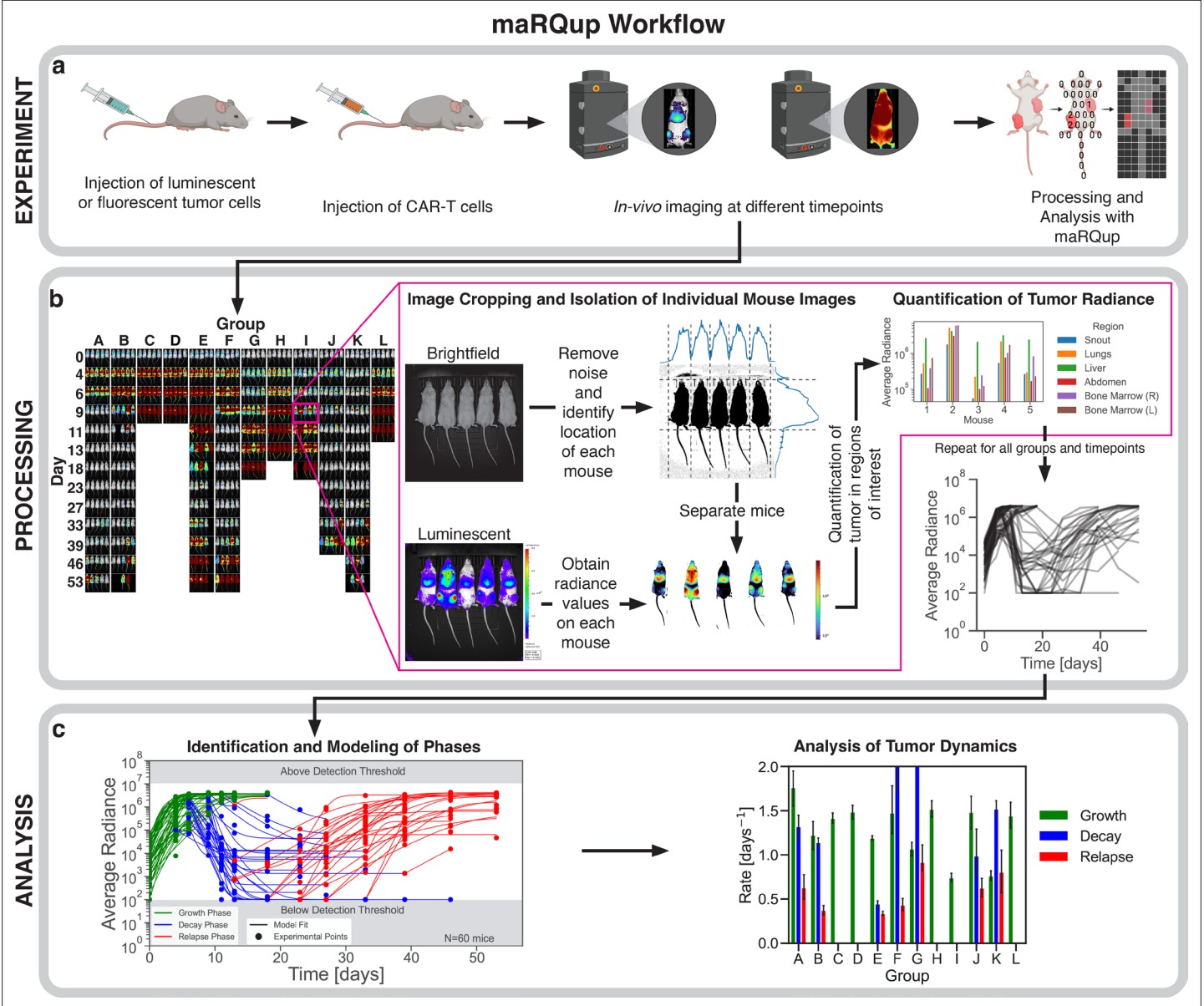

**Figure 1.** Pipeline detailing the workflow of maRQup. (**a**) A schematic of the in vivo experimental setup used to evaluate the efficacy of CAR-T cell immunotherapy experiments. NALM-6 leukemia cells expressing luciferase were injected via the tail vein of mice followed by the adoptive transfer of CAR-T cells (IV) and the evaluation of tumor growth using a Xenogen IVIS Lumina, generating the output raw images for the maRQup pipeline. (**b**) Processing steps to quantify tumor radiance at each timepoint for all mice in an experiment. IVIS output images are grouped by experimental condition (columns) and across multiple timepoints (rows) as indicated. Each output image contains a Brightfield (top) and Luminescent (bottom) image. Individual mice are then separated from the group image, and all mice images are aligned via image processing. Tumor burdens in different anatomical regions are quantified, and the tumor burden over time is calculated. (**c**) Modeling, parametrization, and analysis of tumor behavior. Different phases of the tumor response dynamics—growth, decay, and relapse—are identified for each mouse. Each phase is modeled using differential equations, allowing for the rates of tumor growth in each phase to be compared across mice and experimental conditions.

for each mouse in units of photons per second per cm² per steradian per pixel was calculated and used as a quantification of tumor burden. maRQup also allows the user to select a custom region of interest to calculate tumor burden across all mice. Further details about these steps can be found in the Supplemental Methods Appendix 1.

## Dataset generation

Using our maRQup pipeline, we accrued a dataset consisting of 1060 mice from 37 separate experiments conducted by nine different researchers. Information on the CAR-T cell constructs (extracellular binding domain and intracellular costimulatory domain), CAR-T cell doses, target tumors, and tumor doses for each experiment were appended to our dataset along with the measurements of tumor dynamics.

Of the 1060 mice, almost all (n=908) were injected with an initial load of $0.8–1.0 \times 10^6$ tumor cells. NALM6, an acute lymphoblastic B cell leukemia (B-ALL) cell line, was the most common tumor studied (n=872). Tumors were treated with CAR-T cells targeting either CD19 and/or CD22 cell surface receptors (n=715). CD19-targeting CARs with either a 4-1BB (KYMRIAH-like, tisa-cel) or CD28 costimulatory domain (YESCARTA-like, axi-cel) were used in 382 and 209 mice, respectively. A CD22-targeting CAR with a 4-1BB costimulatory domain, granted FDA breakthrough therapy designation based on its efficacy in children and young adults with relapsed/refractory B-ALL (*Fry et al., 2018*), was used in 74 mice. An additional 55 mice were treated with a bicistronic CD19/CD22-targeting CAR containing CD28 and 4-1BB costimulatory domains, respectively, that are currently being assessed in a phase 1/2 clinical trial (*Shalabi et al., 2022*). Mock transduced T cells (n=143) and no T cells (n=91) were used as controls. CAR-T cells were also evaluated against other B-ALL cell lines in 106 mice, and these mice were only included in the dataset to aid in the development of maRQup and to ensure its generalizability.

The dose of adoptively transferred CAR-T cell was another variable: 126 mice were treated with $<1 \times 10^6$ CAR-T cells, 148 mice were treated with $1 \times 10^6$ CAR-T cells, 42 mice were treated with $1.5 \times 10^6$ CAR-T cells, 411 mice were treated with $2 \times 10^6$ CAR-T cells, and 99 mice were treated with $>2 \times 10^6$ CAR-T cells. The distribution of these experimental conditions are summarized in *Appendix 1—figure 2*.

## Classification of tumor responses to CAR-T cells

From our processed experiments, we identified three phases of tumor dynamics: tumor growth, tumor decay, and tumor relapse. However, not all tumors progressed through all three of these phases. Thus, we used our simple automatic classification of tumor response to CAR-T cell therapy (see section 'Classifying mouse outcomes to CAR-T cell therapies as a function of quantitative tumor dynamics') with five different categories: (G) growth only; (G,D) growth and decay; (G,D,R) growth, decay, and relapse; (D) decay only; and (D,R) decay and relapse (*Appendix 1—figure 2*).

Of the 1060 mice, 602 mice had tumors that initially grew from day 0. Of these mice with initial tumor growth, the tumors then decayed in 228 mice and did not decay in 374 mice. 218 of the 374 mice with no tumor decay (58.3%) represented control mice that were not treated with CAR-T cells. 167 mice had tumors that grew and decayed but eventually relapsed. Of the mice with tumors that decayed initially following CAR-T treatment (n=458, 43.2% of all mice), 355 mice had tumors that relapsed (77.5%). Only 103 mice had tumor that only decayed and never exhibited a growth (or relapse) phase. Hence, despite the tight range of immunotherapeutic parameters, we observed large variability in both tumor growth and eradication (*Appendix 1—figure 2d and e*).

## Comparison of 4-1BB and CD28 costimulatory domains in CD19-targeting CAR-T cells

We first focused our analysis to assess the effect of costimulatory domains in CD19 CAR-T cells harboring the same CD19-targeting FMC63 scFv: anti-CD19.4-1BB (KYMRIAH-like, tisa-cel) and anti-CD19.CD28 (YESCARTA-like, axi-cel) CAR-T cells (*Figure 2a*). These two CAR-T cell constructs have been approved by the FDA as treatments for B-lineage acute lymphoblastic leukemia and lymphoma (*Braendstrup et al., 2020*). These CAR-T cells feature different costimulatory domains and while they are both efficacious, previous publications have highlighted their variable cytotoxic activities and persistence, both in preclinical studies (*Zhao et al., 2015*; *Kawalekar et al., 2016*) and in patients (*Bachy et al., 2022*; *Jacobson et al., 2024*; *Gagelmann et al., 2024*).

In the preclinical settings evaluated here, we found the overall tumor behavior for all mice treated with either CD19.4-1BB CAR-T cells or CD19.CD28 CAR-T cells to be indeed statistically different (*Figure 2b–f*). Of NSG mice with NALM6 tumors that were treated with CD19.4-1BB CAR-T cells (n=377), 43.2% of tumors continued to grow initially even in the presence of CAR-T cells. However,

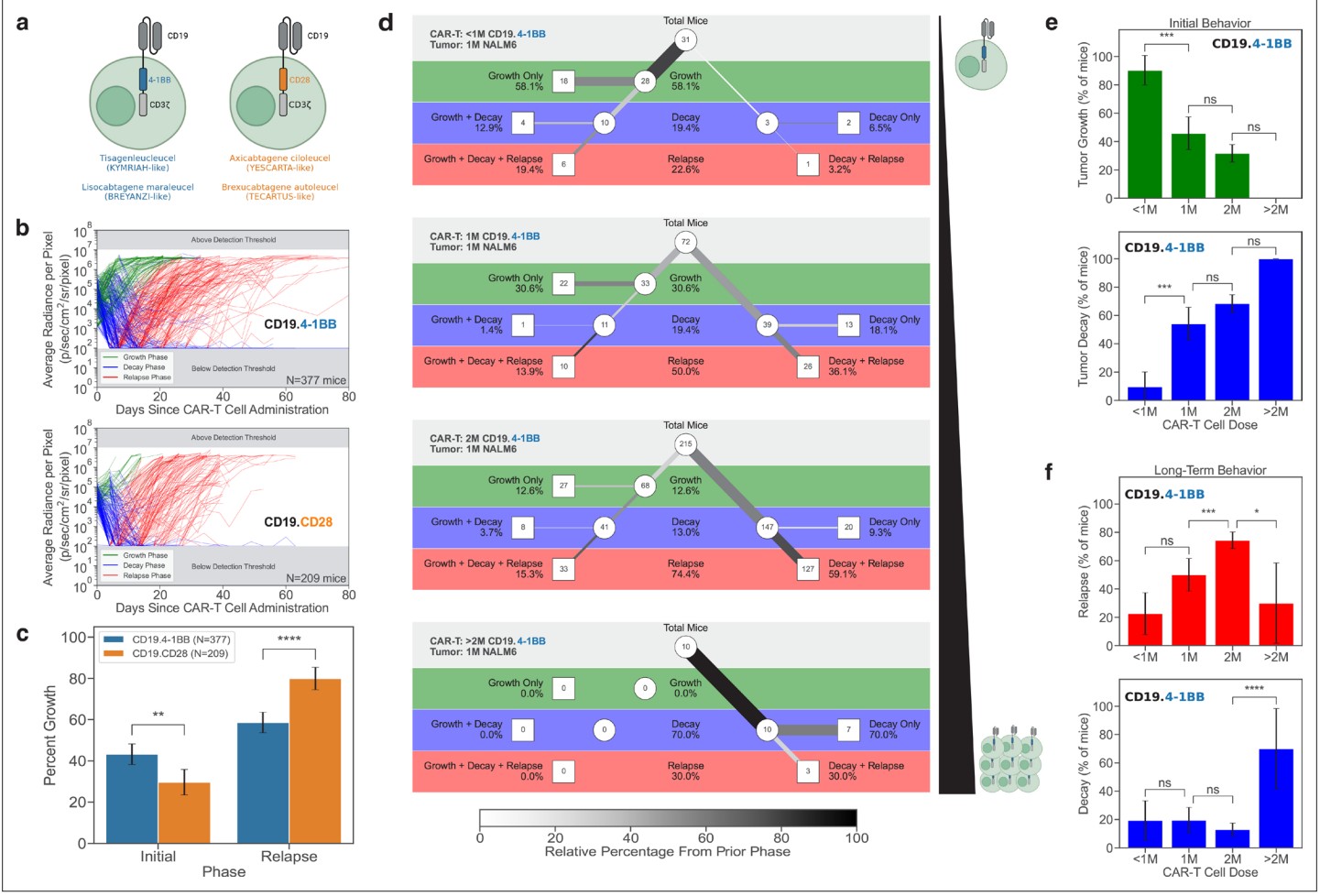

**Figure 2.** Impact of CAR-T cell constructs and doses on in vivo dynamics. (**a**) Schematic showing the extracellular and intracellular components of T cells harboring the CD19.4-1BB (left, blue) and the CD19.CD28 (right, orange) CAR constructs. (**b**) Overall tumor response curves for all mice treated with CD19.4-1BB CAR-T cells (top) and CD19.CD28 CAR-T cells (bottom). (**c**) The percentages of mice treated with CD19.4-1BB (blue) or CD19.CD28 (orange) CAR-T cells that exhibit an initial tumor growth (left) *vs.* those that relapse (right). (**d**) Tree diagrams showing tumor behavior in mice treated with increasing doses of CD19.4-1BB CAR-T cells. (**e**) The percentages of mice demonstrating an initial growth (top, green) as compared to a decay (bottom, blue) of NALM6 leukemia cells presented as a function of the dose of CD19.4-1BB CAR-T cells (<1, 1, 2, or >2 million). (**f**) The percentages of leukemia-bearing mice that exhibit a final phase of relapse (top, red) as compared to decay (bottom, blue) presented as a function of the dose of CD19.4-1BB CAR-T cells.

only 29.7% of mice with NALM6 tumors treated with the CD19.CD28 CAR-T cells (n=209) continued to grow in the presence of CAR-T cells. Thus, once the CD19.CD28 CAR-T cells were injected into the mice, they were more likely than the CD19.4-1BB CAR-T cells to immediately initiate tumor killing and prevent tumor growth. NALM6 tumors treated with CD19.4-1BB CAR-T cells were still able to continue growing, suggesting that CD19.4-1BB CAR-T cells are either slower to home to tumor and/or slower to induce cytotoxicity against NALM6 leukemia. (*Figure 2c*).

Although CD19.CD28 CAR-T cells were initially better at killing NALM6 leukemia, 79.9% of mice eventually relapsed. While there was still significant relapse in mice treated with CD19.4-1BB CAR-T cells, only 58.6% of mice relapsed (*Figure 2c*). These observations, which were not previously reported in previous preclinical mouse experiments, were made statistically significant due to the very large size of the cohorts of mice processed through our maRQup pipeline. This is critical as our preclinical mouse data—demonstrating a more rapid cytotoxicity of CD19.CD28 CAR-T cells and a sustained persistence of CD19.4-1BB CAR-T cells—is in accord with clinical trial outcomes showing higher efficacy of the former in diffuse large B-cell lymphoma and prolonged persistence of the latter (*Maude et al., 2014*; *Maude et al., 2018*; *Cheng et al., 2018*; *Aamir et al., 2021*).

## Effects of CD19.4-1BB CAR-T cell dose on in vivo cytotoxicity against NALM6 leukemia

We next investigated the impact of different doses of CAR-T cells on tumor eradication. We focused on the largest subset of mice with the same initial tumor burden ($1 \times 10^6$ NALM6) treated with the same CD19.4-1BB CAR-T cells (n=349 mice). As expected, increasing the dose of CAR-T cells from $<1 \times 10^6$ cells to $>2 \times 10^6$ cells resulted in an increased initial killing of the tumor and limited tumor growth (*Figure 2d–e*).

When the dose of CAR-T cells was $<1 \times 10^6$ CAR-T cells, leukemia growth initially continued in 90.3% (28/31) of mice. In 18 of these mice (64.3%), the leukemia was never controlled by the CAR-T cells and the mice rapidly succumbed. Increasing the dose to $1 \times 10^6$ CAR-T cells resulted in a higher initial tumor control, with 54.2% (39/72) of tumors regressing immediately following treatment. Doubling the dose to $2 \times 10^6$ CAR-T cells further increased efficacy to 68.4% (147/215), and at doses of $>2 \times 10^6$ CAR-T cells, 100% (10/10) of mice exhibited tumor regression (*Figure 2d and e*).

Although higher doses of CAR-T cells led to a more significant initial tumor killing, it is interesting to note that the frequency of tumor relapse also increased. At doses of $<1 \times 10^6$ and $1 \times 10^6$ CAR-T cells, 22.6% (7/31) and 50.0% (36/72) of mice eventually relapsed with leukemia, respectively, but at a dose of $2 \times 10^6$ CAR-T cells, there was a significantly higher frequency of relapse with 74.4% (160/215) of mice succumbing to leukemia ($p<0.001$). In contrast, at higher doses of $>2 \times 10^6$ CAR-T cells, leukemic growth was better controlled with only 30% (3/10) of mice relapsing (*Figure 2d–f*, $p<0.05$).

These results suggest an ideal dosing regimen that optimizes the trade-off between the highest level of initial tumor killing and the lowest level of tumor relapse. Our analysis also highlights the potential for a maximal immediate tumor killing to be associated with a higher level of relapse, at least in the context of an NSG mouse model. The observations from our maRQup pipeline reveal trade-offs in short-term and long-term CAR-T cell cytotoxicities that will be important to investigate further.

## Parametrizing tumor dynamics following the adoptive transfer of CAR-T cells

We sought to parametrize the dynamics of tumor growth in the NSG mouse model (*Figure 3c*). The data from each phase (growth, decay, and relapse) for all mice were fitted using our piecewise dynamical model as described in section 'Modeling each dynamic phase'. We found that our model could fit all three phases with residuals within the experimental noise (*Figure 3a and b*). For the growth phase, 99.9% of the predicted data were within one order of magnitude of the experimental data (which has a range of five orders of magnitude). Similarly, 99.5% and 97.7% of data points for the decay and relapse phases, respectively, were within one order of magnitude of the experimental data (*Figure 3b*).

## Tumor growth is slower during the relapse phase than the initial phase

In order to compare the rate of tumor growth during the relapse phase to the rate of tumor growth during the initial growth phase, we examined only mice with tumors that grew from the start, exhibited some control by CAR-T cells, and eventually relapsed (G,D,R). Control mice that did not receive CAR-T cells were excluded. A total of 335 mice were engrafted with NALM6 leukemia that initially grew after CAR-T cell administration. 156 of these mice exhibited leukemic relapse (46.6%). The rate of tumor growth during relapse was significantly slower than the rate of tumor growth during the initial phase for 81.4% of these mice (n=127) (*Figure 3d*), highlighting the selective pressure that CAR-T cell cytotoxicity applies to the leukemia. Future studies will be needed to characterize these selective pressures at both the molecular and cellular levels.

## Rates of initial leukemia growth predict the long-term behavior

We report the distribution of the initial growth rates for all 335 mice with NALM6 leukemia that continue to grow upon CAR-T cell treatment to be bimodal. Two classes of responses to CAR-T cells could be distinguished from the initial tumor growth phase: 170 mice (50.7%) had slowgrowing tumors ($k_{Growth} \leq 0.85$ days$^{-1}$), while 165 mice (49.3%) had fast-growing tumors ($k_{Growth} > 0.85$ days$^{-1}$) (*Figure 3e*).

Such dynamic differences in the initial phase of the preclinical experiments resulted in long-term differences. Slower growing leukemia were more likely to escape complete killing by CAR-T cells

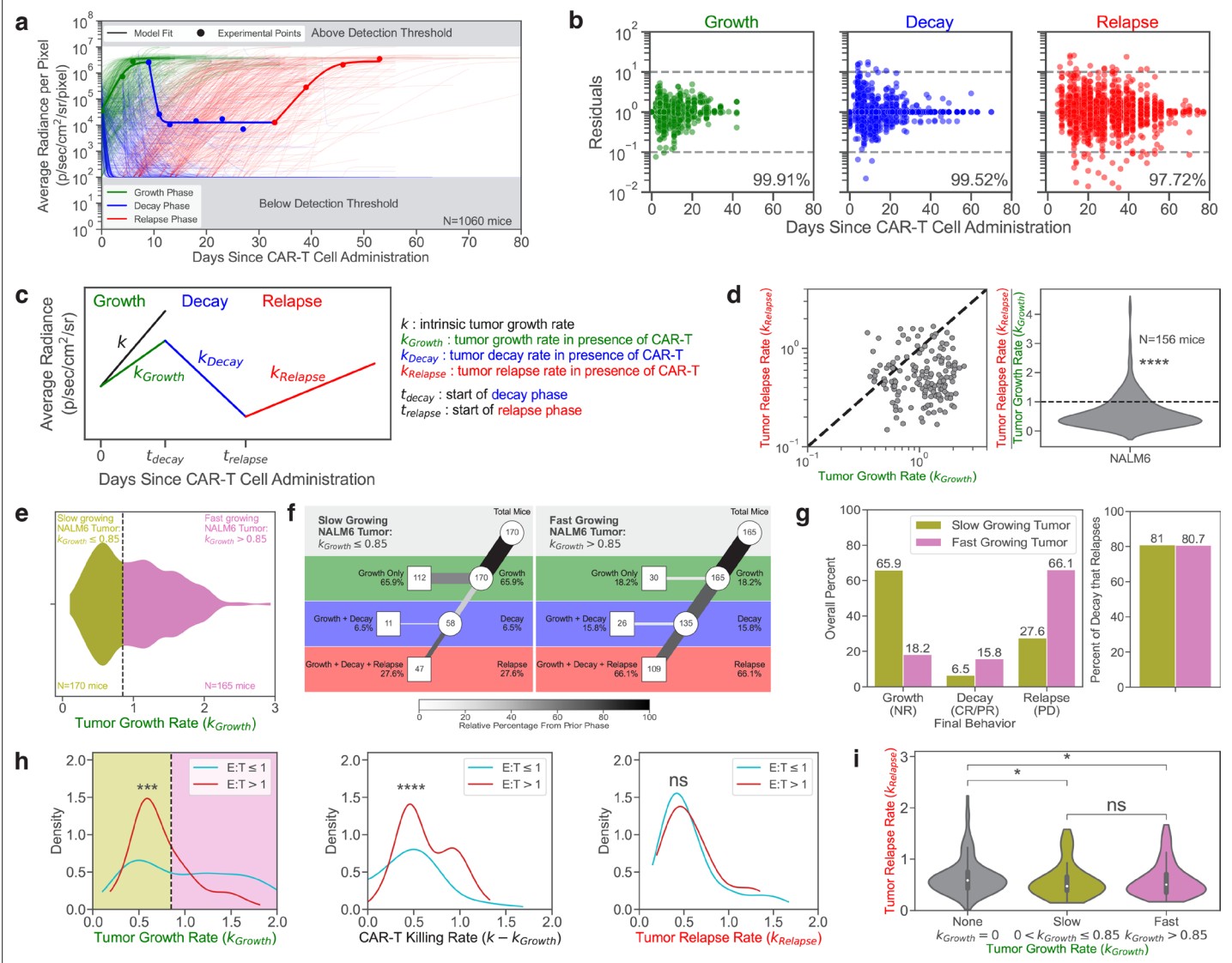

**Figure 3.** Parametrization of tumor response to CAR-T cell therapy. (**a**) Tumor dynamics for all 1060 mice in the dataset. Model calculations are shown as curves for all mice. A representative example is highlighted, with the experimental data shown as points. The growth, decay, and relapse phases are labeled in green, blue, and red, respectively. (**b**) Residuals in $log_{10}$-scale are shown for each timepoint in each phase. The percent of modeled points within one order of magnitude in either the positive or negative direction of the experimental points (dotted gray lines; $10^{-1}$–$10^1$) are labeled for each phase. (**c**) Diagram displaying the parameters used to model the tumor dynamics. (**d**) Comparison of the growth rate of NALM6 leukemia in the growth phase ($k_{Growth}$) and in the relapse phase ($k_{Relapse}$) in mice that exhibited tumor growth, decay, and relapse. (**e**) Tumor growth rates can be divided into slow-growing (gold) and fast-growing populations (purple). The vertical dotted line at 0.85 days$^{-1}$ represents the division of the two populations. (**f**) Tree diagram for the slow-growing (left) and fast-growing tumor populations (right). (**g**) (left) Percentage of mice harboring slow or fast-growing tumor populations as a function of their last phase being growth (NR), decay (CR/PR), or relapse (PD). (right) The percentages of mice that exhibit tumor decay and then relapse under conditions where the tumors exhibit slow (gold) or fast (purple) initial growth. (**h**) Kernel density estimate plots of the distribution of the growth rate of tumors in the growth phase ($k_{Growth}$) (left), the rate of tumor killing by CAR-T cells ($k - k_{Growth}$) (middle), and the growth rate of tumors in the relapse phase ($k_{Relapse}$) (right). Mice were classified into two groups of Effector to Target Ratios (E:T) depending on if the mice were treated with more (red) or fewer/equal (blue) CAR-T cells than tumor cells. The division of the slow (gold) and fast (purple) growing tumor populations, as determined from e is shown. (**i**) Violin plots showing the growth rate of tumors during the relapse phase ($k_{Relapse}$) as stratified by the initial behavior of the tumor: initial decay ($k_{Growth} = 0$), slow initial growth ($0 < k_{Growth} \leq 0.85$), or rapid initial growth ($k_{Growth} > 0.85$). All rates have units of days$^{-1}$.

with 65.9% (112/170) of these leukemia never regressing. In contrast, only 18.2% (30/165) of fast-growing leukemia escaped killing. Indeed, we detected regression in 81.8% (135/165) of fast-growing tumors vs. 34.1% (58/170) of slow-growing tumors. However, the faster growing tumors resulted in significantly more relapse than the slower growing tumors, with 66.1% (109/165) of tumors eventually

relapsing. Interestingly though, the likelihood of relapse after regression was similar for the fast and slow-growing tumors (80.7% (109/135) and 81.0% (47/58), respectively). These results suggest that slow-growing leukemia may avoid being killed by CAR-T cells. While this may limit short-term expansion, it can lead to continued long-term tumor growth. In contrast, faster growing tumors are initially killed by CAR-T cells, but following this phase, the likelihood of relapse is similar (*Figure 3f and g*). Hence our maRQup analysis reveals a quantitative paradox that leukemia that are initially better controlled by CAR-T cells may evade CAR-T cell killing in the long run.

## Effects of CAR-T:Tumor cell numbers on in vivo growth and killing

We also explored the effects of different ratios of CAR-T to tumor cells on in vivo growth and killing rates. We split our dataset into two categories depending on whether mice were treated with more (CAR-T:Tumor ≥ 1) or fewer (CAR-T:Tumor < 1) CAR-T cells than the original number of injected tumor cells. The distributions of tumor growth rates (initial and relapse) and CAR-T cell killing rates for these two groups are shown in *Figure 3h*. The initial growth rate of tumors was significantly faster when fewer CAR-T cells were injected (*Figure 3h*, left, p<0.001), suggesting that the effects of the CAR-T cell dose contributes to the bimodal distribution of initial tumor growth rates seen in *Figure 3e*. Consequently, the rate of killing of the tumors by the CAR-T cells, as calculated as the difference in tumor growth rate in the presence and absence of CAR-T cells, was significantly greater when a higher number of CAR-T cells were adoptively transferred into tumor-bearing mice (*Figure 3h*, middle, p<0.0001).

Although the rate of initial tumor growth was modulated by the ratio of CAR-T cells to tumor cells, there was no significant difference in the growth rate of tumors during the relapse phase (*Figure 3h*, right) suggesting that tumor growth during the relapse phase is independent of the CAR-T cell dose. The distribution of tumor growth rates during the relapse phase was dependent on the classification of overall tumor behavior—growth, decay, and relapse ($k_{Growth} > 0$); and decay and relapse ($k_{Growth} = 0$). The 'growth, decay, and relapse' population could further be stratified by the rate of initial tumor growth (i.e., fast vs. slow, as shown in *Figure 3e*). We found that there was no difference in the rate of tumor growth during the relapse phase among tumors that were initially fast or slow growing. However, the rate of tumor growth during the relapse phase was significantly faster in tumors that regressed immediately following CAR-T cell administration as compared to tumors that continued growing upon adoptive transfer of CAR-T (*Figure 3i*, p<0.05).

## Tracking of tumor burden across anatomical regions following adoptive transfer of CAR-T cells

Each pixel of the mouse tumor radiance image consists of three different layers of data: the tumor radiance value, a boolean value to distinguish the mouse from the background, and a brightfield image value (*Appendix 1—figure 4a*). As described in section 'maRQup pipeline', all images were processed and aligned so that anatomical regions were located in the same area across all images. *Appendix 1—figure 4g* shows the standard deviation of the boolean values of the merged images as well as the average radiance, mouse, and brightfield values across all 7534 images. From the average radiance image, all pixel values within the same row were averaged, and a principal component analysis was performed. The first four principal components accounted for greater than 90% of the overall variability (*Appendix 1—figure 5a*, left). The same process was performed with the column pixels (*Appendix 1—figure 5a*, right). By using the x-intercepts of the loading curves for row principal components 2 and 3, and column principal components 4 and 5, we were able to reliably demarcate nine distinct anatomical regions (*Figure 4a*; *Appendix 1—figure 5a*, middle/left). After quantifying tumor radiance within different anatomical regions, the tumor burden within these regions could be monitored separately over time. Intriguingly, the dynamics of tumor growth appeared to vary per region (representative example shown in *Figure 4b*). We therefore systematically characterized the tumor growth patterns per region across all 872 NALM6 leukemia-bearing mice in our dataset (*Figure 4c*, left) and found a remarkable level of divergence in the growth patterns of leukemic cells in individual regions compared to the overall mouse tumor burden (*Figure 4c*, right and *Appendix 1—figure 5b–d*): up to 40% of analyzed mice showed different growth patterns in an individual anatomical region compared to their overall tumor burden. Characterizing this divergence per anatomical region and growth pattern revealed that the snout and bone marrow regions show significantly different

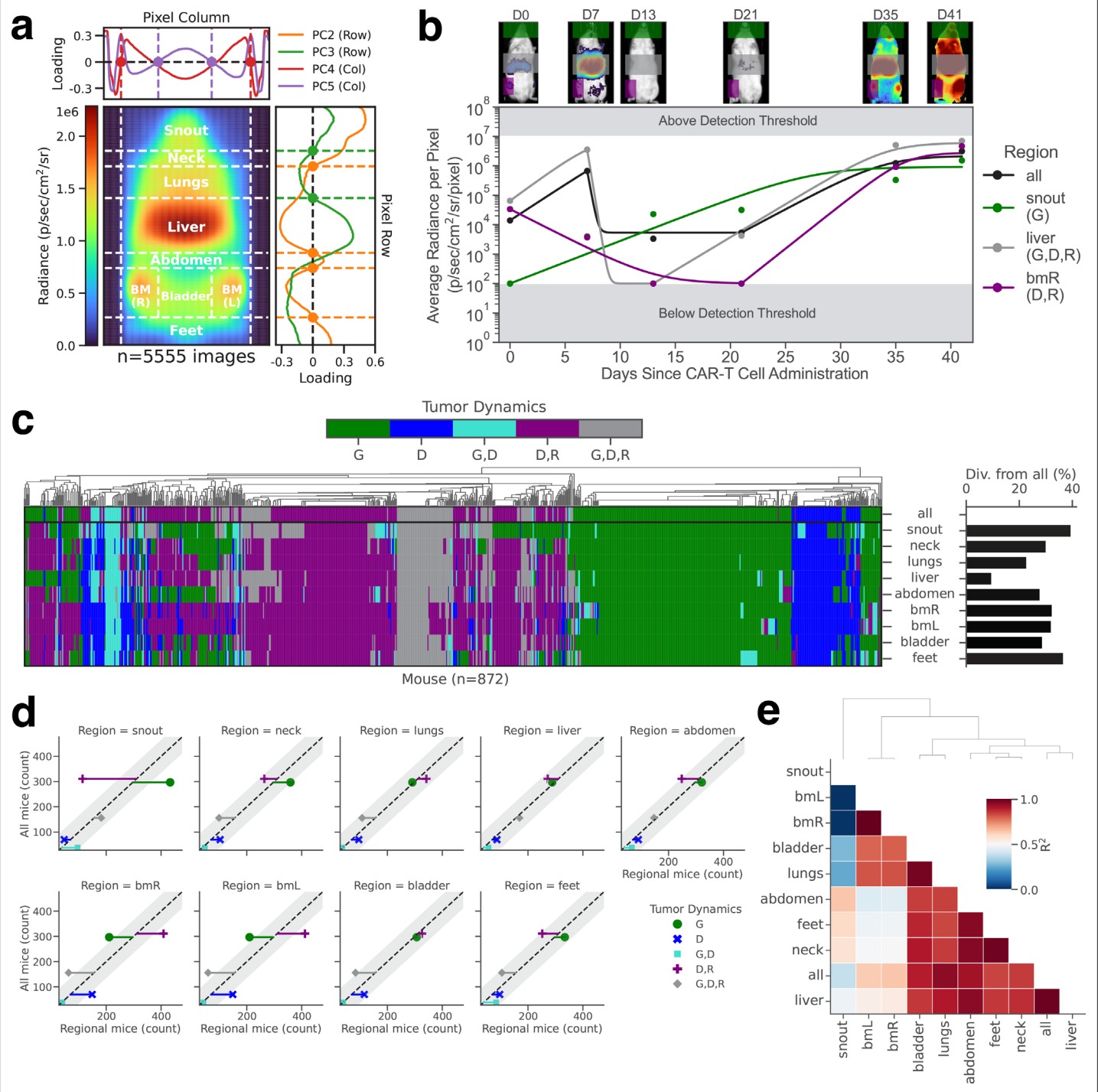

**Figure 4.** Characterizing spatial differences in tumor progression dynamics. (**a**) Zeros of four principal components measured across row-averaged and column-averaged mouse image data were used to demarcate nine distinct spatial regions. (**b**) Images (top) and quantified radiance dynamics (bottom) both globally (all) and per region are shown for a representative mouse. (**c**) Characterization of tumor growth dynamics in all mice engrafted with NALM6 leukemia (left) and the percentages of mice where tumor dynamics in a given region diverge from the dynamics evaluated from the overall total body radiance (right). The 'bmR' region is defined as the region containing the bone marrow of the mouse's right leg; the 'bmL' region is defined as the region containing the bone marrow of the mouse's left leg. (**d**) Scatter plots of mice with a given overall tumor behavior as compared to the classified tumor behavior within each specific region. Residuals from the identity line (dashed diagonal line) with magnitudes outside the 95% confidence intervals (gray bands) indicate a significant divergence of a region-specific growth pattern as compared to the overall growth pattern. (**e**) Correlation coefficients for all pairwise combinations of overall tumor growth and region-specific tumor growth.

tumor growth dynamics than the overall tumor growth across the whole mouse (*Figure 4d*). Leukemia in the snout was much more likely to only grow over time (and never decay) in mice exhibiting bulk leukemia regression prior to relapse, indicating that snout-localized leukemic cells were more resistant to CAR-T cell immunotherapy. In direct contrast, leukemia in the bone marrow was more likely to regress and relapse even when the overall tumor burden in the mice would only grow or grow, decay, and relapse, suggesting that bone marrow-localized tumors can be more easily killed by CAR-T cells. This sharp deviation between the two regions could also be observed by the extremely low correlation of growth dynamics between snout and bone marrow tumors (*Figure 4e*).

Thus, our spatiotemporal analysis of tumor growth dynamics derived from maRQup indicates that in many mice, the tumor in the bone marrow is eradicated by CAR-T cells and that a few tumor cells in the snout are resistant to killing by CAR-T cells. These tumor cells then circulate, resulting in global tumor relapse. These results suggest that CAR-T cells are either unable to access leukemia in the snout or that a subset of CAR-T cell resistant-leukemia inherently localize to the snout region. We presented these observations to illustrate the power of our maRQup pipeline to derive novel quantitative observations that call for further molecular and cellular analyses.

## Discussion

Here we have used maRQup to assemble a large dataset of preclinical experiments yielding important insights about the response dynamics of B cell acute lymphoblastic leukemia tumor cells to different CAR-T cell doses and constructs. From our data, we quantified differences between CD19 CAR-T cells harboring a 4-1BB as compared to a CD28 costimulatory signaling domain in in vivo murine experiments. CD19.CD28 CAR-T cells initially were more effective at killing the tumor and preventing initial tumor growth. However, these CAR-T cells were associated with a high frequency of tumor relapse, consistent with limited long-term activation and/or exhaustion. Conversely, adoptive transfer of CD19.4-1BB CAR-T cells was associated with a significantly greater frequency of initial tumor growth, but of note, there were fewer relapses as compared to mice treated with CD19.CD28 CAR-T cells. These results reveal that CD19.4-1BB CAR-T cells require longer kinetics for initial homing and/or cytotoxic function, but they retain killing potential for more sustained periods of time. Importantly, while these types of conclusions were not able to be drawn from preclinical mice experiments prior to the maRQup analyses presented here, they are in accord with clinical findings in patients: CD19.CD28 CAR-T cells exhibit high efficacy in the context of lymphoma, but CD19.4-1BB CAR-T cells persist for significantly longer periods of time (*Maude et al., 2014*; *Maude et al., 2018*; *Cheng et al., 2018*; *Aamir et al., 2021*; *Long et al., 2015*; *Lee et al., 2015*; *Shah and Fry, 2019*).

The potential for a time-controlled administration, resulting in a relay from exhausted CAR-T cells to CAR-T cells that function at a later time point can be of much interest (*Salazar-Cavazos and Altan-Bonnet, 2022*). However, future experiments investigating the impact of such a strategy on severe adverse effects such as cytokine release syndrome are needed. Our maRQup pipeline will aid in the optimization of this type of CAR-T cell regimen, facilitating preclinical testing and analysis.

Using maRQup, we have also documented the impact of CAR-T cell dose on tumor dynamics in a preclinical setting. As expected, administration of a higher number of CAR-T cells is associated with an increase in initial tumor killing. However, it was of interest to note that higher doses of CAR-T cells did not prevent relapse. We conjecture, that at higher doses, CAR-T cells do eradicate tumors too efficiently and rapidly, limiting the release of cytokines (e.g., IL-2) necessary to prolong and sustain CAR-T cell cytotoxicity. However, these data were observed in the context of NSG mice wherein homeostatic murine cytokines do not cross-react with human cytokine receptors; it will therefore be of interest to determine whether the data reported here are model-specific or whether relapse might be lower in humanized mice models engineered to express human cytokines such as MISTRG mice (*Strowig et al., 2011*; *Ivic et al., 2017*). It is, however, notable that at very high doses of CAR-T cells, a complete eradication of tumor at early time points prevents relapse. In patients, the trade-off between efficacy and severe adverse effects such as cytokine release syndrome and on-target/off-tumor effects needs to be carefully evaluated. This trade-off can be assessed as a 'Pareto Front' (*Shoval et al., 2012*) - balancing the need for tumor killing, cytokine secretion by CAR-T cells activated by tumor antigens, persistence, and lack of exhaustion to prevent long-term tumor relapse. Further detailed modeling of such Pareto optimization can be performed using the quantitative parameters derived from maRQup analysis.

Additionally, the image processing from maRQup enables for the spatiotemporal evaluation of tumor dynamics following treatment with CAR-T cells. Applying this pipeline to our dataset, we identified the snout as an anatomical region that has different tumor behavior as compared to the overall response of the tumor. Tumor in the snout appears to be more resistant to CAR-T cell therapy, either due to the snout being an immune-privileged site or a region where resistant tumor cells (i.e., inherently resistant or resistant by down-regulating the target antigen) localize. Multiple case reports have been published describing similar observations in patients who have had B cell acute lymphoblastic leukemia tumor relapse in the mandibles (*Bakathir and Al-Hamdani, 2009*; *Benson et al., 2007*; *Penafort et al., 2024*).

We anticipate that systematic application of maRQup will allow investigators and clinicians to rigorously, comparatively, and rapidly optimize CAR-T cell design and dose regimens, leading to improved cancer immunotherapies for patients.

## Materials and methods

### In vivo evaluation of CAR-T cell immunotherapeutic potential

We followed standard protocols to assess the in vivo efficacy of CAR-T cells against cancer cells (e.g., patient-derived cell lines [*Fåne et al., 2022*] or xenografts of cancer cell lines [*Liu et al., 2023*]). In this study, we analyzed data from a large collection of experiments (*Figure 1a*) where NALM-6 cells (ATCC CRL-3273: a patient-derived acute lymphoblastic leukemia cell line), engineered to express luciferase, were adoptively-transferred into humanized immunodeficient NSG mice (NOD.Cg-*Prkdc*$^{scid}$*Il2rg*$^{tm1W jl}$/ SzJ; Jackson Laboratories). After a three day engraftment period of the tumor cells, the human CAR-T cells of interest were adoptively transferred by intravenous (IV) injection. All mouse injections were performed by one expert, thus reducing inter-researcher variability. Tumor dynamics were monitored via luciferase-based imaging of each mouse as indicated, using the Xenogen IVIS Lumina machine (Caliper Life Sciences). Mice received an intraperitoneal injection of 3 mg of D-luciferin (Caliper Life Sciences) followed by a 4 min incubation time prior to imaging using fixed settings (e.g., all exposure times were set to 30 s for all mice). Brightfield images of mice were also collected for image registration. All animal experiments were carried out under the protocols approved by the National Cancer Institute's Institutional Animal Care and Use Committee (IACUC protocol PB-027-4G).

### Classifying mouse outcomes to CAR-T cell therapies as a function of quantitative tumor dynamics

For each mouse, tumor radiance was calculated at all timepoints as a measure of responsiveness to CAR-T cells. We observed three qualitatively distinct phases of tumor dynamics upon reviewing our datasets: tumor growth, tumor decay, and tumor relapse. To automatically identify these three phases, consecutive timepoints were compared in a stepwise fashion until a threshold was met to signal the end of one phase and the start of the next phase (*Appendix 1—figure 1a*): (1) The phase of 'tumor growth' ended at the timepoint immediately prior to when tumor radiance was <75% of the tumor radiance detected at the previous timepoint; (2) 'tumor decay' ended at the timepoint immediately prior to a tumor radiance increase of threefold; and (3) 'tumor relapse' was defined as all remaining timepoints (prior to sacrifice or death). Equations were then derived to automatically fit and delineate each phase (*Appendix 1—figure 1b*). The tumor dynamics in the three aforementioned phases for each individual mouse in each experiment were manually inspected to confirm that the automated classification of tumor dynamics, based on the defined thresholds, was satisfactory.

### Modeling each dynamic phase

We modeled the dynamics of each of these three phases using simple differential equations to examine the kinetic rates of tumors in the growth, decay, and relapse phases (*Figure 1c*). Hence, we performed our parametrization of the tumor dynamics using a piecewise approach. These kinetic rates were then compared across mice that were treated with different CAR-T cell therapeutic parameters (e.g., dose or construct).

The tumor growth and relapse phases were modeled as logistic growth:

$$\frac{dT(t)}{dt} = kT(t)\left(1 - \frac{T(t)}{T_\infty}\right)$$

(1)

where $T(t)$ is the tumor radiance at time $t$. The tumor growth rates $k$ [day$^{-1}$] are $k_{Growth}$ for the initial tumor growth rate and $k_{Relapse}$ for the tumor growth rate during the relapse phase. $T_{\infty}$ is the value of tumor at the end of the initial growth or relapse phase (e.g., carrying capacity for the tumor cells in the recipient mice in terms of measured radiance).

The tumor decay phase was modeled as an exponential decay with a baseline tumor burden, $T_B$, and tumor decay rate, $k_{Decay}$ [day$^{-1}$]:

$$\frac{dT(t)}{dt} = -k_{Decay}T(t) \tag{2}$$

Assuming that tumor dynamics consist of all three phases, the equations for each phase can be written as follows:

Growth phase:

$$T\left(t_{i=1...D}\right) = \frac{T_0}{1 + \frac{T_0}{T_{\infty}}\left(e^{+k_{Growth}(t_i)} - 1\right)}e^{+k_{Growth}(t_i)} \tag{3}$$

Decay phase:

$$T\left(t_{i=D...R}\right) = T_B + \left(\frac{T_0 e^{+k_{Growth}(t_{Decay})}}{1 + \frac{T_0}{T_{\infty}}\left(e^{+k_{Growth}(t_{Decay})} - 1\right)} - T_B\right)_{*e^{-k_{Decay}(t_i - t_{Decay})}} \tag{4}$$

Relapse phase:

$$T\left(t_{i=R...N}\right) = \frac{T_{1,Relapse}e^{+k_{Relapse}(t_i - t_{Relapse})}}{1 + \frac{T_{1,Relapse}}{T_{\infty}}\left(e^{+k_{Relapse}(t_i - t_{Relapse})} - 1\right)} \tag{5}$$

where

$$T_{1,Relapse} = T_B + \left(\frac{T_0 e^{+k_{Growth}(t_{Decay})}}{1 + \frac{T_0}{T_{\infty}}\left(e^{+k_{Growth}(t_{Decay})} - 1\right)} - T_B\right)_{*e^{-k_{Decay}(t_{Relapse} - t_{Decay})}} \tag{6}$$

Hence, the entire dynamics of tumor growth, decay, and relapse can be parametrized with eight parameters:

1. $T_0$, the tumor radiance at $t=0$
2. $k_{Growth}$, the rate of tumor growth in the first phase
3. $t_{Decay}$, the time when the tumor mass starts shrinking ($D^{th}$ timepoint)
4. $k_{Decay}$, the rate of tumor disappearance in the decay phase
5. $T_B$, the baseline tumor radiance during the decay phase
6. $t_{Relapse}$, the time when the tumor mass starts growing again ($R^{th}$ timepoint)
7. $k_{Relapse}$, the rate of tumor growth in the relapse phase
8. $T_{\infty}$, the tumor radiance at the carrying capacity (or the end of the growth or relapse phase if the carrying capacity is not yet reached)

The dynamics of tumor responses to CAR-T cell immunotherapies can also consist of one or two phases (e.g., immediate obliteration of tumor growth and/or the absence of tumor relapse). In these cases, tumor dynamics can be modeled with equations *Equation 4* or combined equations *Equation 3* and *Equation 4*. The derivation of the equations for each phase, which include the equations for simpler configurations of tumor response, can be found in Supplemental Information.

## Bayesian inference for dynamic parameters

We fit the experimental tumor radiances using the above equations and obtained values for $k_{Growth}$, $k_{Decay}$, and $k_{Relapse}$. However, we found that there were parameter outliers that most frequently

occurred when there were only two data points in the phase used for fitting (***Appendix 1—figure 3b***). To correct for these, we implemented a fitting routine based on Bayesian inference in order to optimize the fit while constraining the values of these parameters to a realistic range (***Gelman et al., 2017***; ***Gelman et al., 2020***).

For the initial growth rate, $k_{Growth}$, the prior distribution of the fitting parameter was determined by removing outliers where $k_{Growth} > 3$ days$^{-1}$. After removal of the outliers, the prior population used had a mean $k_{Growth} = 0.84$ days$^{-1}$ and standard deviation of 0.43 days$^{-1}$.

For the growth rate during relapse, $k_{Relapse}$, the prior distribution of the fitting parameter was determined by removing outliers where $k_{Relapse} > 3$ days$^{-1}$ as well. After removal of the outliers, the prior population used had a mean $k_{Relapse} = 0.60$ days$^{-1}$ and standard deviation of 0.32 days$^{-1}$.

Note that the fit for the decay rate, $k_{Decay}$, yielded a bimodal population. Thus, we could not derive reliable Bayesian priors for the tumor decay phase. Such simple fitting procedure reduced the complex dynamics of tumor response to CAR-T cell treatment to a small set of quantitative parameters, allowing for additional statistical analysis.

***Appendix 1—figure 3*** shows the Bayesian Priors method of removing and refitting outliers.

## Materials availability

All materials and reagents will be available by contacting the corresponding authors.

## Acknowledgements

We thank members of the Taylor lab for their help in sharing data and designing and test-running the processing pipeline. We thank members of the Altan-Bonnet lab for comments and discussions. Funding This work was partially supported by the intramural research program of the NCI and by a NCI Flex Synergy Award to NT and GA-B.

## Additional information

### Funding

| Funder | Grant reference number | Author |
| --- | --- | --- |
| NCI Intermural Research Program | | Naomi Taylor<br>Grégoire Y Altan-Bonnet |
| NCI Flex Synergy Award | | Naomi Taylor<br>Grégoire Y Altan-Bonnet |

The funders had no role in study design, data collection and interpretation, or the decision to submit the work for publication.

### Author contributions

Adam L Kenet, Conceptualization, Data curation, Software, Formal analysis, Validation, Investigation, Visualization, Methodology, Writing – original draft, Project administration, Writing – review and editing; Sooraj Achar, Conceptualization, Data curation, Software, Formal analysis, Validation, Investigation, Visualization, Methodology, Writing – original draft, Writing – review and editing; Alka Dwivedi, Marie Pouzolles, Haying Qin, Christopher Chien, Resources, Data curation, Investigation; John Buckley, Resources, Data curation; Naomi Taylor, Resources, Data curation, Formal analysis, Supervision, Funding acquisition, Investigation, Writing – original draft, Project administration, Writing – review and editing; Grégoire Y Altan-Bonnet, Conceptualization, Resources, Data curation, Software, Formal analysis, Supervision, Funding acquisition, Validation, Investigation, Visualization, Methodology, Writing – original draft, Project administration, Writing – review and editing

### Author ORCIDs

Grégoire Y Altan-Bonnet ⓘ https://orcid.org/0000-0002-9602-8133

### Ethics

All animal experiments were carried out under the protocols approved by the National Cancer Institute's Institutional Animal Care and Use Committee (IACUC protocol PB-027-4-G).

Reviewer #1 (Public review): https://doi.org/10.7554/eLife.106470.3.sa1
Reviewer #3 (Public review): https://doi.org/10.7554/eLife.106470.3.sa2
Author response https://doi.org/10.7554/eLife.106470.3.sa3

---

## Additional files

### Supplementary files

MDAR checklist

### Data availability

All data are available in the following repository: https://github.com/soorajachar/radianceQuantifier/tree/main/radianceQuantifier/data_for_paper. An updated version of maRQup is available in the following Github repository https://github.com/soorajachar/radianceQuantifier (copy archives at *Kenet and Achar, 2026*).

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

## Appendix 1

### Supplemental methods

#### Image processing

For each timepoint, the Xenogen IVIS Lumina outputs two files of interest: brightfield and luminescent images. The brightfield images are standard visible-light images of the mice. On the brightfield images, the mice appear white on a black background. The luminescent images are infrared luminescence images that register with the localization of the tumor within the mice. From the brightfield images, we used white-pixel thresholding to automatically identify the location of each individual mouse on the image and cropped the images to isolate each individual mouse from the group image. Since the luminescent image was captured concomitantly as the brightfield image, we applied the cropping of the brightfield image to the luminescent image for each respective mouse. Then, only pixels that were on the mouse as determined from the brightfield image were used to calculate tumor burden from the radiance of the luminescent image. This step was necessary in the case that the tumor burden was so large that it saturated the detector and spilled-over onto the background of the image. *Figure 1b* summarizes these steps.

#### Cropping the signal from the mouse tails

The random positioning of a mouse's tail resulted in large variability in the size (length and width) of each image. Thus, the tail of each mouse was automatically cropped out of the images to streamline processing. In order to automatically crop each mouse's tail, brightfield values were summed across each one-pixel high horizontal plane of the mouse. The horizontal plane with the maximum summed value was used to divide each mouse into superior and inferior parts. Then, the horizontal plane in the inferior section at 80% of the maximum summed value was used as an additional threshold. Only those planes inferior to this threshold were used for further analysis. The summed values of these planes were then filtered using a Savitzky–Golay filter (*Schafer, 2011*) with a window length of 51 pixels and a polynomial order of 2 to remove noise in the data. The start of each mouse's tail was identified using the KneeLocator algorithm from the *kneed* Python toolkit. All horizontal planes inferior to the plane identified by this algorithm were removed as these planes correspond to the tail of the mouse. *Appendix 1—figure 4b and c* shows an example of this tail cropping method.

After the tails were cropped, images were scaled to normalize the mouse length. This scaling ensures that each anatomical region of interest lies in the same horizontal slice across all the mice. The Python packages *imutils* and *OpenCV* were used to scale the images vertically to a height of 345 pixels while keeping the aspect ratio unchanged. Pixels were interpolated using the bilinear interpolation method (*INTER_LINEAR*) from *OpenCV*.

#### Removal of slanted images

Next, an ordinary least squares regression was performed on the (x,y) coordinates of the mouse pixels for each image to define the spinal axis of the mouse. The angle that formed between the line of best fit and the vertical (arctangent of the slope of the line of best fit) was used to identify images with slanted mice (*Appendix 1—figure 4d*). These slanted images were typically wider than the average image. Thus, any image with an angle greater than six degrees (e.g., because of misplacement during image acquisition) was removed (*Appendix 1—figure 4e*). Only 1.2% (91/7633) of all images were removed according to this threshold. Including these images unnecessarily increased the background padding for all images, significantly increased the computational effort required for stacking, and lessened the reliability of our method. An additional eight images were abnormally wide due to incorrect image cropping and were also removed (*Appendix 1—figure 4f*). The remaining 7534 images were padded with –1, 0, and the minimum (i.e., background) brightfield value in the radiance, mouse pixels, and brightfield layers, respectively, to a width of 185 pixels. This processing allowed for the alignment of the mice and the stacking of all the images for future analyses. All 7534 images were stacked, and the standard deviation of the mouse pixels (1 if the pixel is on the mouse, 0 if the pixel is background) was calculated to confirm the image processing steps properly align all the mice (*Appendix 1—figure 4g*).

## Quantification of tumor radiance

Tumor burden was then calculated from the images for all mice at each timepoint. The mouse pixel layer of each image was used to determine which pixels were on the mouse and may contain tumor radiance. In some experiments, when the tumor burden of one mouse saturates the detector, some radiance can spillover to the background of the image. Thus, only pixels that are on the mice as determined from the mouse pixel values were used to calculate tumor burden. The average radiance per pixel [photons per second per cm² per steradian per pixel] for each mouse was calculated and used as a quantification of tumor burden.

## Supplemental modeling

### Derivation of modeling equations

Let $N$ be the number of timepoints in each respective phase, with $N > 0$. Solving the differential equation, *Equation 1*, for $i = 1 \ldots N$ with the initial condition $T(t_{i=1}) = T_1$ yields

$$T\left(t_{i=1\ldots N}\right) = \frac{T_1}{1 + \frac{T_1}{T_\infty}\left(e^{+k(t_i - t_1)} - 1\right)} e^{+k(t_i - t_1)} \tag{SI.1}$$

For the tumor growth phase, $t_{i=1} = 0$ and $T\left(t_{i=1} = 0\right) = T_{1,Growth} = T_0$. Substituting into *Equation SI.1* yields *Equation 3*:

$$T\left(t_{i=1\ldots N}\right) = \frac{T_0}{1 + \frac{T_0}{T_\infty}\left(e^{+k_{Growth}(t_i)} - 1\right)} e^{+k_{Growth}(t_i)} \tag{SI.2}$$

For the tumor relapse phase, $t_{i=1} = t_{Relapse} > 0$ and $T\left(t_{Relapse} > 0\right) = T_{1,Relapse}$. Substituting into *Equation SI.1* yields *Equation 5*:

$$T\left(t_{i=1\ldots N}\right) = \frac{T_{1,Relapse}}{1 + \frac{T_{1,Relapse}}{T_\infty}\left(e^{+k_{Relapse}(t_i - t_{Relapse})} - 1\right)} e^{+k_{Relapse}(t_i - t_{Relapse})} \tag{SI.3}$$

Solving the differential equation, *Equation 2*, with initial condition $T\left(t_{i=1} = t_{Decay} \geq 0\right) = T_{1,Decay}$ yields

$$T\left(t_{i=1\ldots N}\right) = T_B + \left(T_{1,Decay} - T_B\right) e^{-k_{Decay}(t_i - t_{Decay})} \tag{SI.4}$$

If $t_{Decay} = 0$, then there is no initial tumor growth phase. Therefore, $t_{i=1} = t_{Decay}$ and

$$T\left(t_{Decay} = 0\right) = T_{1,Decay} = T_0 \tag{SI.5}$$

If $t_{Decay} > 0$, then there is an initial tumor growth phase. Therefore, $t_{i=1} = t_{Decay} = t_{end,Growth}$ and from *Equation SI.2*,

$$T\left(t_{Decay} > 0\right) = T_{1,Decay} = \frac{T_0}{1 + \frac{T_0}{T_\infty}\left(e^{+k_{Growth}(t_{Decay})} - 1\right)} e^{+k_{Growth}(t_{Decay})} \tag{SI.6}$$

Thus, substituting *Equation SI.6* into *Equation SI.4* yields *Equation 4*:

$$T\left(t_{i=1\ldots N}\right) = T_B + \left(\frac{T_0 e^{+k_{Growth}(t_{Decay})}}{1 + \frac{T_0}{T_\infty}\left(e^{+k_{Growth}(t_{Decay})} - 1\right)} - T_B\right) e^{-k_{Decay}(t_i - t_{Decay})} \tag{SI.7}$$

At the start of the relapse phase, $t_{Relapse} = t_{end,Decay}$, and from *Equation SI.4*,

$$T\left(t_{Relapse}\right) = T_{1,Relapse} = T_B + \left(T\left(t_{Decay}\right) - T_B\right) e^{-k_{Decay}(t_{Relapse} - t_{Decay})} \tag{SI.8}$$

In the scenario where there is no initial growth phase ($t_{Decay} = 0$; *Equation SI.5*), then *Equation SI.8* simplifies to

$$T\left(t_{Relapse}\right) = T_{1,Relapse} = T_B + \left(T_0 - T_B\right) e^{-k_{Decay}\left(t_{Relapse} - t_{Decay}\right)} \tag{SI.9}$$

In the scenario where there is an initial growth phase ($t_{Decay} > 0$; **Equation SI.6**), then **Equation SI.8** simplifies to **Equation 6**:

$$T\left(t_{Relapse}\right) = T_{1,Relapse} = T_B + \left(\frac{T_0 e^{+k_{Growth}\left(t_{Decay}\right)}}{1 + \frac{T_0}{T_\infty}\left(e^{+k_{Growth}\left(t_{Decay}\right)} - 1\right)} - T_B\right) e^{-k_{Decay}\left(t_{Relapse} - t_{Decay}\right)} \tag{SI.10}$$

Now the equations for each phase can be rewritten in terms of only the following parameters: $T_0$, $T_\infty$, $T_B$, $k_{Growth}$, $k_{Decay}$, $k_{Relapse}$, $t_{Decay}$, and $t_{Relapse}$ (see **Equations 3–6**).

### Determination of carrying capacity

If the radiance of the last two timepoints of the growth or relapse phase were within a factor of 10 of each other, it was assumed that the tumor cells reached their carrying capacity, which thus was estimated as the average of the radiance of the last two timepoints. However, if the tumor was still growing at the end of the growth or relapse phase, it was assumed that the carrying capacity was not yet reached, and the carrying capacity was set to the default radiance of $10^7$ photons per second per cm$^2$ per steradian per pixel. This value is the global maximum, or saturation, of tumor radiance for each mouse, as determined by examining the carrying capacity across the numerous other mice which had tumors that reached the carrying capacity. This maximum tumor radiance was consistently around $10^7$ photons per second per cm$^2$ per steradian per pixel with little variability, indicating that it is the saturation limit of the IVIS imaging machine.

## Supplemental figures

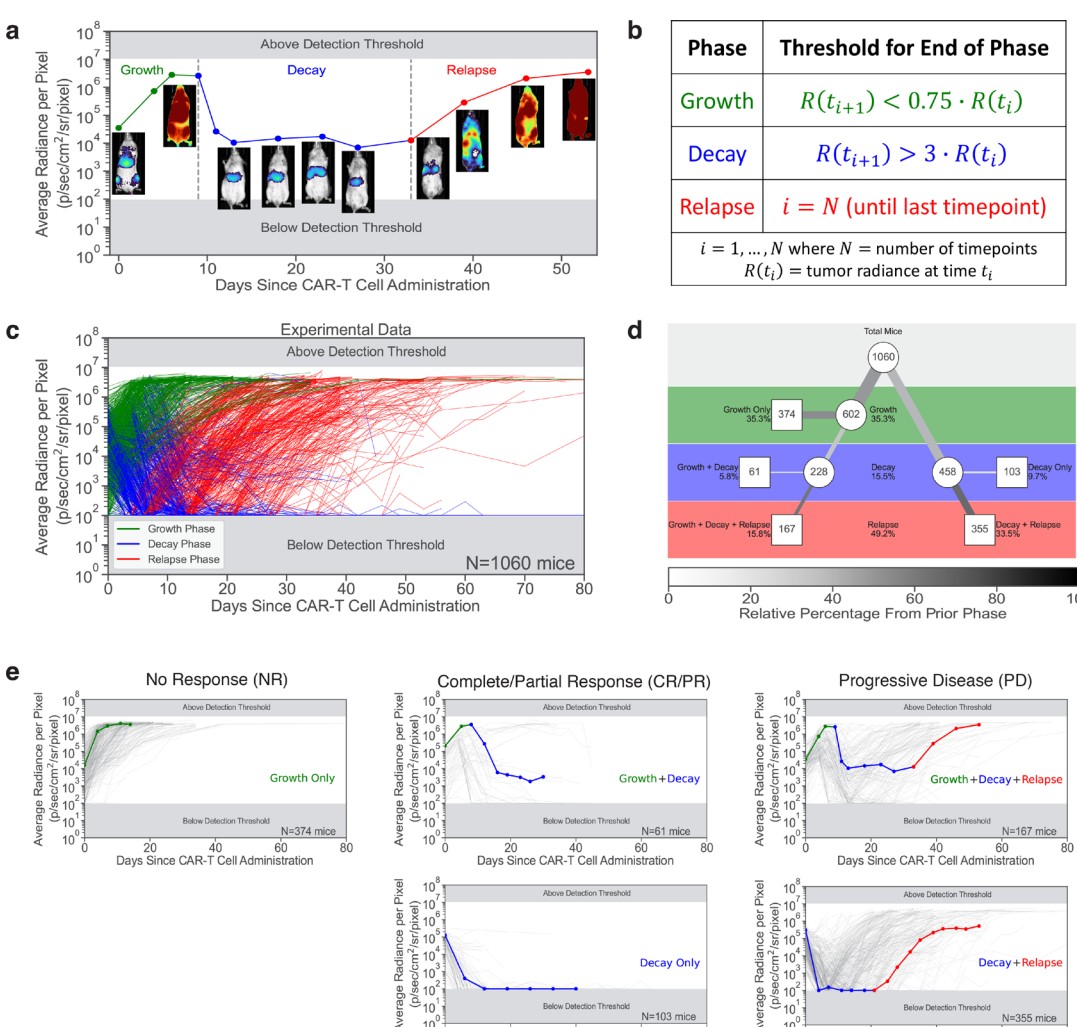

**Appendix 1—figure 1.** maRQup classification of tumor response dynamics. (**a**) Representative example of the tumor dynamics showing growth (green), decay (blue), and relapse (red) phases for a single mouse. Tumor radiance images are shown below each corresponding timepoint. (**b**) Equations used to determine the timepoint for the end of each phase. (**c**) Experimental tumor dynamics for all 1,060 mice in the dataset showing the three different phases. (**d**) Tree diagram detailing the tumor behavior for all 1060 mice. Each row corresponds to a different phase (i.e., growth, decay, or relapse), and each node (circle) indicates the number of mice with tumor in each phase. Each endpoint (square) indicates the number of mice with the respective tumor dynamics. The thickness of each line denotes the overall percentage of mice that pass through that path, while the darkness of each line denotes the relative percentage of mice that move from the previous node to the next node or endpoint. (**e**) Classification of tumor dynamics into three categories: no response (NR); complete/partial response (CR/PR); and progressive disease (PD). Tumor growth curves for all mice in the respective categories are shown. A representative tumor curve is shown on each plot with the corresponding phases identified.

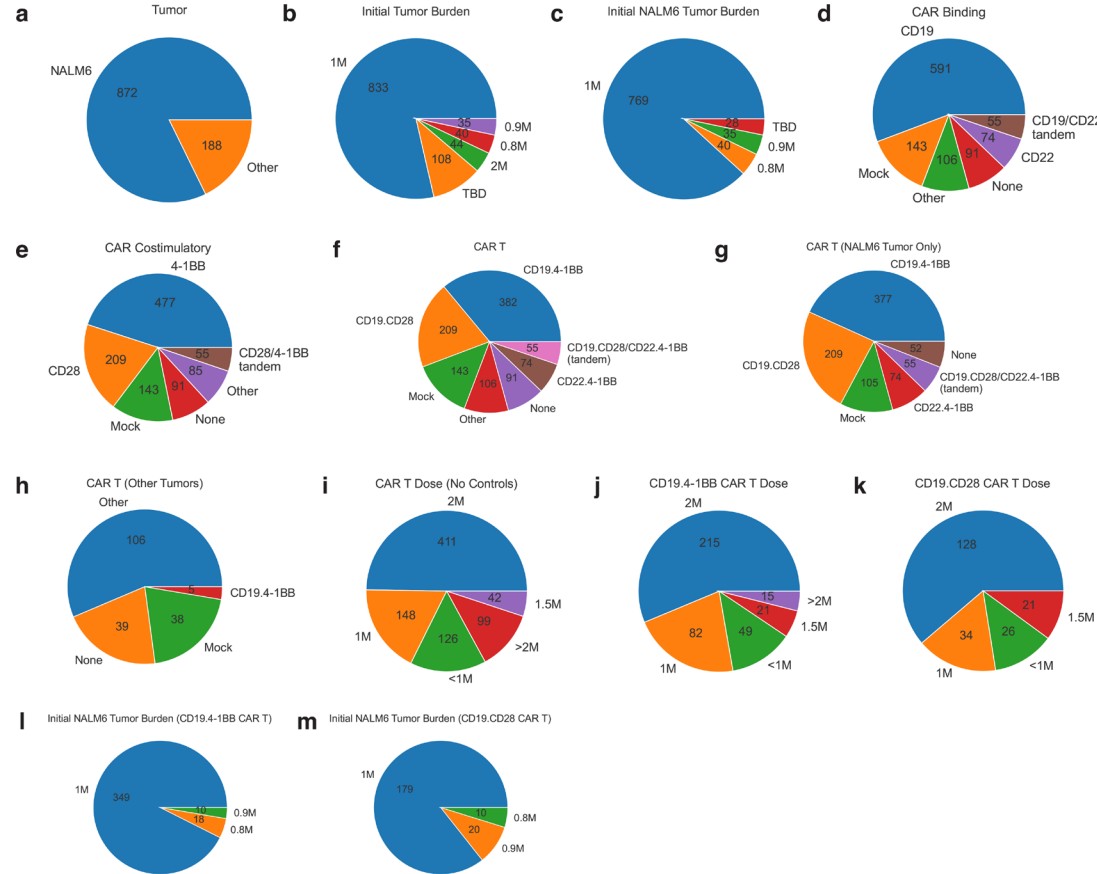

**Appendix 1—figure 2.** Summary of different experimental conditions in the dataset. (**a**) Different tumors in the dataset. (**b**) Initial dose of all tumors injected into the mice. (**c**) Initial dose of NALM6 tumors injected into the mice. (**d**) Different binding domains for CAR-T cells. (**e**) Different costimulatory domains for CAR-T cells. (**f**) Different CAR-T cell constructs used to target all tumors in the dataset. (**g**) Different CAR-T cell constructs used to target NALM6 tumors only. (**h**) Different CAR-T cell constructs used to target non-NALM6 tumors. (**i**) Doses of CAR-T cells (not including mock/control conditions). (**j**) Doses of the CD19.4-1BB CAR-T cells. (**k**) Doses of the CD19.CD28 CAR-T cells. (**l**) Initial dose of NALM6 tumors that were treated with CD19.4-1BB CAR-T cells. (**m**) Initial dose of NALM6 tumors that were treated with CD19.CD28 CAR-T cells.

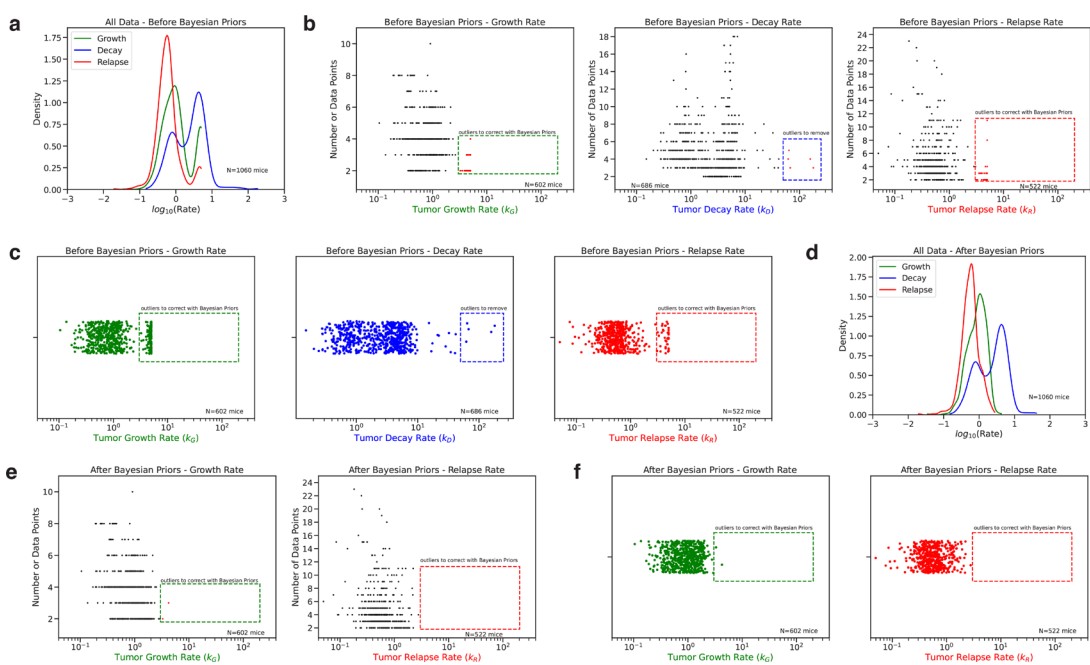

**Appendix 1—figure 3.** Bayesian Priors methods. (**a**) Distributions of the rate of tumor growth in the growth phase (green), the rate of tumor decay in the decay phase (blue), and the rate of tumor growth in the relapse phase (red) prior to Bayesian Priors correction. (**b**) Total number of data points per phase as a function of the tumor growth/decay rate before Bayesian Priors correction. Outlier data points are shown in red in the dotted box for each phase. (**c**) Distribution of rates before Bayesian Priors correction for each phase with outliers identified in the dotted box for each phase. (**d**) Distributions of the rate of tumor growth in the growth phase (green), the rate of tumor decay in the decay phase (blue), and the rate of tumor growth in the relapse phase (red) after Bayesian Priors correction. (**e**) Total number of data points per phase as a function of the tumor growth rate in the growth and relapse phases after Bayesian Priors correction. Outlier data points are shown in red in the dotted box for each phase. The decay phase is not shown since Bayesian Priors was not applied to this phase. (**f**) Distribution of rates after Bayesian Priors correction for the growth and relapse phases with remaining outliers identified in the dotted box for each phase.

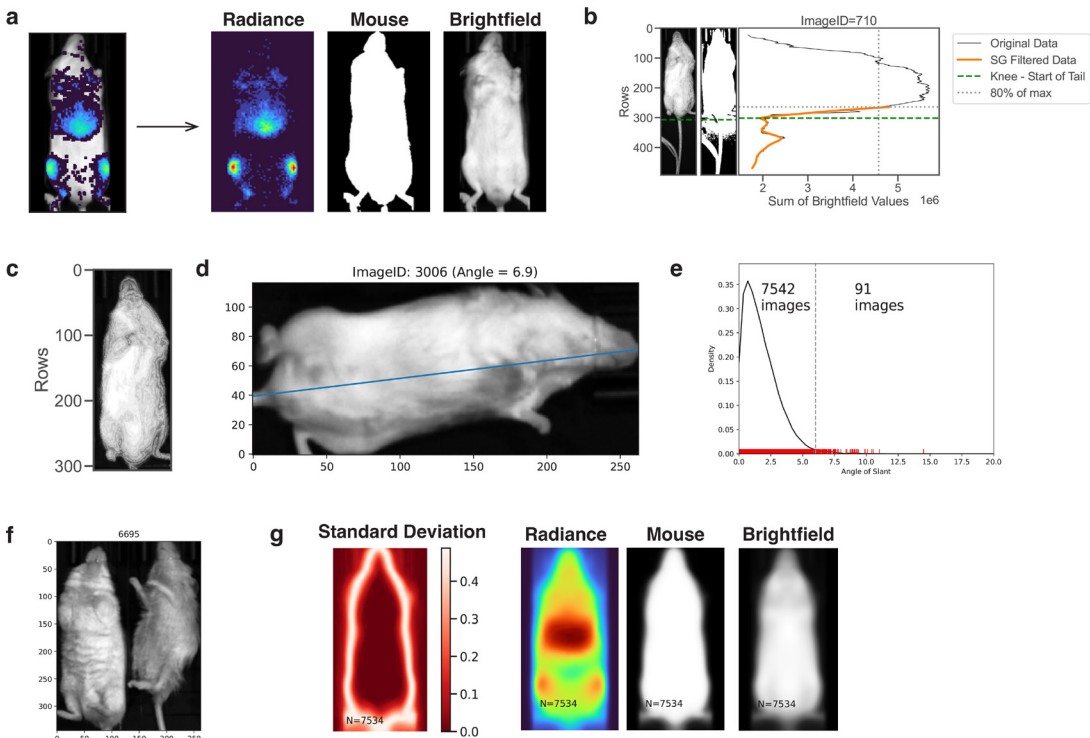

**Appendix 1—figure 4.** Image processing methods. (**a**) Each composite IVIS image consists of a radiance image (left), a binary mouse pixel image (middle), and a brightfield image (right). (**b**) Tail cropping method. Dotted green line denotes the row where the start of the tail is identified via a Savitzky–Golay filter (SG, orange curve) and KneeLocator algorithm. (**c**) Representative mouse image following tail cropping. (**d**) Representative image of a slanted mouse. The ordinary least squares regression line of the (x,y) pixel coordinates is shown in blue. (**e**) Distribution of the angle of slant for all 7633 images in the dataset. Each red line represents one image. Images were removed from analysis if the angle of slant was greater than 6°. (**f**) Representative example of one of eight images that were removed due to incorrect image cropping (e.g., multiple mice in one image). (**g**) Composite images (i.e., standard deviation of the merged mouse pixel boolean images, average of the merged radiance images, average of the merged mouse pixel images, and average of the merged brightfield images) after all 7534 images were merged following image processing.

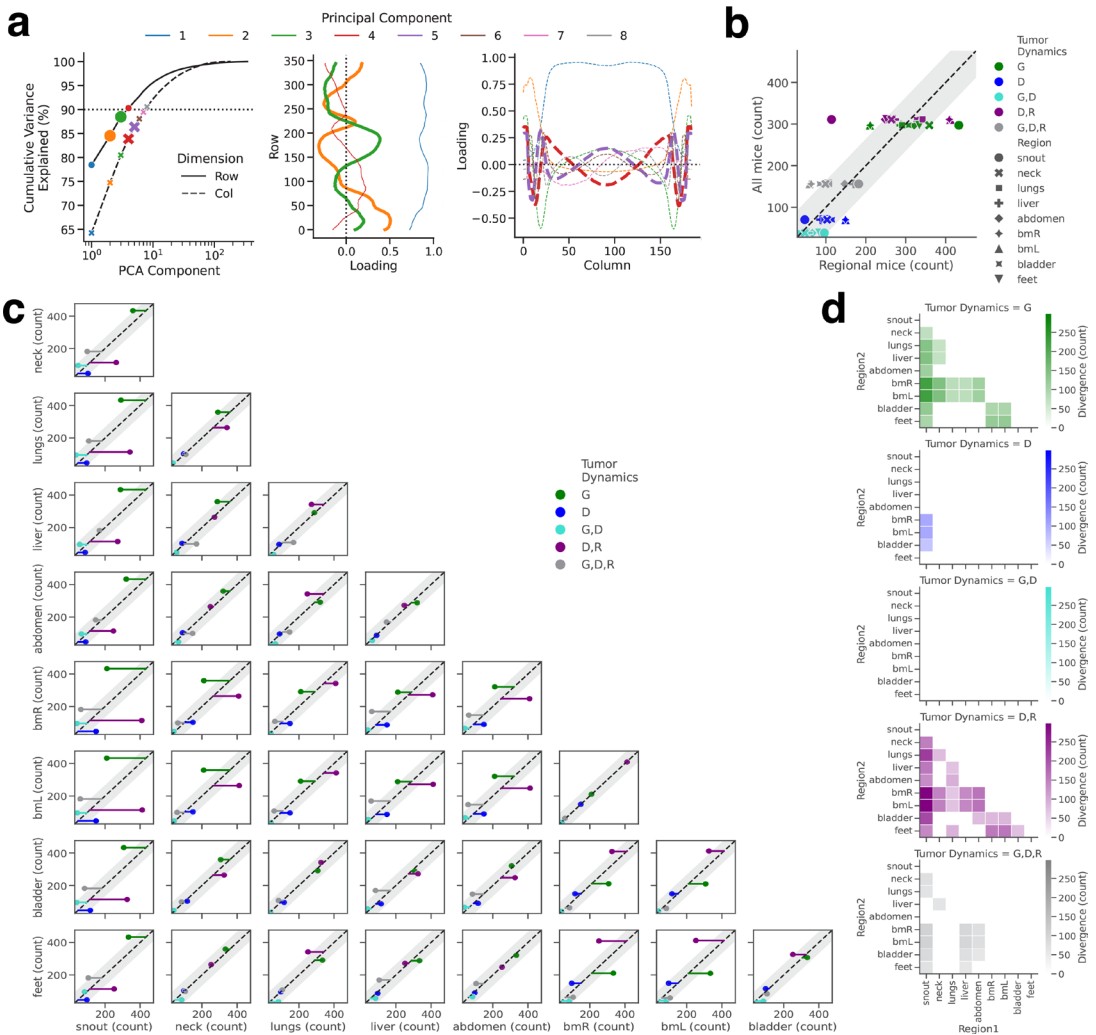

**Appendix 1—figure 5.** Spatial partitioning. (**a**) (left) Cumulative variance of principal components for the principal component analysis performed in the row-dimension (solid line) or column-dimension. The principal components that cumulatively explain greater than 90% of the variance are shown in different colors. (Middle) Values across all rows of the first four principal components from the row-wise principal component analysis. Principal component 2 (orange) and principal component 3 (green) are shown in bold and were used to identify anatomical regions of interest. (Right) Values across all columns of the first eight principal components from the column-wise principal component analysis. Principal component 4 (red) and principal component 5 (purple) are shown in bold and were used to identify anatomical regions of interest. (**b**) Number of mice with region-specific behavior compared to overall tumor behavior for the nine previously identified anatomical regions. Residuals from the identity line (dashed diagonal line) with magnitudes outside of 95% confidence intervals (gray bands) indicate a significant divergence of a region-specific growth pattern from the overall growth pattern. (**c**) Pairwise comparisons of region-specific tumor behavior. (**d**) Heatmaps showing the number of mice that have different tumor behavior across each region shown for each of the five classified tumor behaviors (i.e., growth only [G; green]; decay only [D; blue]; growth and decay [G,D; teal]; decay and relapse [D,R; purple]; and growth, decay, and relapse [G,D,R; gray]).

