## [Editor Report · eLife Assessment]

The authors developed a **fundamental** computational method, which is intended to automatically process bioluminescence imaging-derived tumor images across anatomical regions and over time. This allows quantitative analysis of such data, and the authors applied it to describe the spatiotemporal distribution of tumour cells in response to CD19-targeted CAR-T cells that contained either CD28 or 4-1BB costimulatory domains. Some operational limitations were identified, which relate to the pipeline's reliance on predefined regions of interest instead of aligning signal sites with anatomical information, scaling, and limitations in taking animal pose into account. Overall, the authors provide **compelling** evidence for the functionality of their computational approach towards automated analysis of bioluminescence imaging data, while applying it to a current topic of wide interest in cell therapy research.

---

## [Referee Report · Reviewer #1 (Public review)]

Summary:

This paper presents maRQup a Python pipeline for automating the quantitative analysis of preclinical cancer immunotherapy experiments using bioluminescent imaging in mice. maRQup processes images to quantify tumor burden over time and across anatomical regions, enabling large-scale analysis of over 1,000 mice. The study uses this tool to compare different CAR-T cell constructs and doses, identifying differences in initial tumor control and relapse rates, particularly noting that CD19.CD28 CAR-T cells show faster initial killing but higher relapse compared to CD19.4-1BB CAR-T cells. Furthermore, maRQup facilitates the spatiotemporal analysis of tumor dynamics, revealing differences in growth patterns based on anatomical location, such as the snout exhibiting more resistance to treatment than bone marrow.

Strengths:

(1) The maRQup pipeline enables the automatic processing of a large dataset of over 1,000 mice, providing investigators with a rapid and efficient method for analyzing extensive bioluminescent tumor image data.

(2) Through image processing steps like tail removal and vertical scaling, maRQup normalizes mouse dimensions to facilitate the alignment of anatomical regions across images. This process enables the reliable demarcation of nine distinct anatomical regions within each mouse image, serving as a basis for spatiotemporal analysis of tumor burden within these consistent regions by quantifying average radiance per pixel.

Weaknesses:

(1) While the pipeline aims to standardize images for regional assessment, the reliance on scaling primarily along the vertical axis after tail removal may introduce limitations to the quantitative robustness of the anatomically defined regions. This approach does not account for potential non-linear growth across dimensions in animals of different ages or sizes, which could result in relative stretching or shrinking of subjects compared to an average reference.

(2) Furthermore, despite excluding severely slanted images, the pipeline does not fully normalize for variations in animal pose during image acquisition (e.g., tucked body, leaning). This pose variability not only impacts the precise relative positioning of internal anatomical regions, potentially making their definition based on relative image coordinates more qualitative than truly quantitative for precise regional analysis, but it also means that the bioluminescent light signal from the tumor will not propagate equally to the camera as photons will travel differentially through the tissue. This differing light path through tissues due to variable positioning can introduce large variability in the measured radiance that was not accounted for in the analysis algorithm. Achieving more robust anatomical and quantitative normalization might require methods that control animal posture using a rigid structure during imaging.

Comments on revisions:

(1) Clarification of 2D Analysis. We strongly recommend that the authors explicitly define maRQup as a 2D spatiotemporal analysis technique. Since optical imaging quantification is inherently dependent on tissue type and signal depth, characterizing this as a 3D or volumetric method without tomographic correction is inaccurate. Please precede "spatiotemporal" with "2D" throughout the text to ensure precision regarding the method's capabilities.

(2) Data Validation and Scaling in Supplemental Figure g currently lacks the units necessary to support the assertion.

Non-Uniform Growth: The authors' method implies that mouse growth is linear and uniform in all directions (isotropic). However, murine growth is not akin to the inflation of a balloon; animals elongate and widen at different rates. The current scaling does not account for these physiological non-linearities.

Pose Variability: The scaling approach appears to neglect significant variability in animal positioning. Even under anesthesia, animal pose is rarely identical across subjects or time points.

Requirement for Evidence: Without quantitative data, there appears to be significant differences between the individual images and the merged image. If the authors assert that this is a "classical setting" where mouse positioning is 100% consistent and growth curves are identical in multiple dimensions, please provide specific references that validate these assumptions. Otherwise, the scaling must be corrected to account for anisotropic growth and pose differences or stated that scaling was only based on one dimension.

(3) Methodology of Spatial Regions The manuscript does not currently indicate how the nine distinct spatial regions were determined. Please expand the methods section to include the specific segmentation algorithms or anatomical criteria used to define these regions, as this is critical for reproducibility.

---

## [Referee Report · Reviewer #3 (Public review)]

Summary:

The paper "The 1000+ mouse project: large-scale spatiotemporal parametrization and modeling of preclinical cancer immunotherapies" is focused on developing a novel methodology for automatic processing of bioluminescence imaging data. It provides quantitative and statistically robust insights on preclinical experiments that will contribute to optimizing cell-based therapies. There is an enormous demand for such methods and approaches that enable the spatiotemporal evaluation of cell monitoring in large cohorts of experimental animals.

Strengths:

The manuscript is generally well written, and the experiments are scientifically sound. The conclusions reflect the soundness of experimental data. This approach seems to be quite innovative and promising to improve the statistical accuracy of BLI data quantification.

This methodology can be used as a universal quantification tool for BLI data for in vivo assessment of adoptively transferred cells due to the versatility of the technology.

Comments on revisions:

The critiques have been taken care of appropriately.

---

## [Author Response]

The following is the authors’ response to the original reviews.

**Public Reviews:**

**Reviewer #1 (Public review):**
Summary:This paper presents maRQup, a Python pipeline for automating the quantitative analysis of preclinical cancer immunotherapy experiments using bioluminescent imaging in mice. maRQup processes images to quantify tumor burden over time and across anatomical regions, enabling large-scale analysis of over 1,000 mice. The study uses this tool to compare different CAR-T cell constructs and doses, identifying differences in initial tumor control and relapse rates, particularly noting that CD19.CD28 CAR-T cells show faster initial killing but higher relapse compared to CD19.4-1BB CAR-T cells. Furthermore, maRQup facilitates the spatiotemporal analysis of tumor dynamics, revealing differences in growth patterns based on anatomical location, such as the snout exhibiting more resistance to treatment than bone marrow.Strengths:(1) The maRQup pipeline enables the automatic processing of a large dataset of over 1,000 mice, providing investigators with a rapid and efficient method for analyzing extensive bioluminescent tumor image data.(2) Through image processing steps like tail removal and vertical scaling, maRQup normalizes mouse dimensions to facilitate the alignment of anatomical regions across images. This process enables the reliable demarcation of nine distinct anatomical regions within each mouse image, serving as a basis for spatiotemporal analysis of tumor burden within these consistent regions by quantifying average radiance per pixel.Weaknesses:(1) While the pipeline aims to standardize images for regional assessment, the reliance on scaling primarily along the vertical axis after tail removal may introduce limitations to the quantitative robustness of the anatomically defined regions. This approach does not account for potential non-linear growth across dimensions in animals of different ages or sizes, which could result in relative stretching or shrinking of subjects compared to an average reference.

Our answer to this comment is included in the Supplemental Methods. The standard deviation of the mouse pixels was calculated to ensure that the image processing steps did not alter the shape or size of the mice. Such consistency is particularly striking because our dataset was accrued by nine lab members over the last five years, before we conceived and carried out our analysis (c.f., answer to point #2). In fact, it is the very consistency of this IVIS measurement that led us to conceive our pipeline. As seen from Supplemental Figure 4G, there is minimal difference in the shape or size of the mice across 7,534 images. A total of 99 images were removed either due to being too slanted (91/7663, 1.2%) or due to processing errors (8/7633, 0.1%). Also, the vertical scaling was conducted while keeping the aspect ratio unchanged to prevent any non-anatomical scaling. Hence, we did not record any nonlinear growth of the mice that would warrant more convoluted alignment and/or batch correction for our images.

(2) Furthermore, despite excluding severely slanted images, the pipeline does not fully normalize for variations in animal pose during image acquisition (e.g., tucked body, leaning). This pose variability not only impacts the precise relative positioning of internal anatomical regions, potentially making their definition based on relative image coordinates more qualitative than truly quantitative for precise regional analysis, but it also means that the bioluminescent light signal from the tumor will not propagate equally to the camera, as photons will travel differentially through the tissue. This differing light path through tissues due to variable positioning can introduce large variability in the measured radiance that was not accounted for in the analysis algorithm. Achieving more robust anatomical and quantitative normalization might require methods that control animal posture using a rigid structure during imaging.

Reviewer #1 is correct that different mouse postures would be an issue when aligning the images and normalizing for size. However, all experiments are conducted for luminescence measurements in the IVIS system (i.e., this requires anesthesia and long integration time for imaging). In our experience and in our 1000+ mouse dataset, we noticed that all experiments (n=37) did place the anesthetized mice in a stretched/elongated position. Of note, these experiments were conducted by nine different researchers who were not instructed on how to place the mice on the machine for ideal image processing, thus showing that the standard protocol of imaging mice on IVIS does not introduce large variations in animal pose during image acquisition. We think the issue raised by Reviewer #1 is moot in the context of classical settings for mouse luminescence imaging.

**Reviewer #2 (Public review):**
Summary:The authors developed a method that automatically processes bioluminescent tumor images for quantitative analysis and used it to describe the spatiotemporal distribution of tumor cells in response to CD19-targeting CAR-T cells, comprising CD28 or 4-1BB costimulatory domains. The conclusion highlights the dependence of tumor decay and relapse on the number of injected cells, the type of cells, and the initial growth rate of tumors (where initial is intended from the first day of therapy). The authors also determined the spatiotemporal analysis of tumor response to CAR T therapy in different regions of the mouse body in a model of acute lymphoblastic leukemia (ALL).Strengths:The analysis is based on a large number of images and accounts for many variables. The results of the analysis largely support their claims that the kinetics of tumor decay and relapse are dependent on the CAR T co-stimulatory domain and number of cells injected and tumor growth rates.Weaknesses:The study does not specify how (a) differences in mouse positioning (and whether they excluded not-aligned mice) and (b) tumor spread at the start of therapy influenced their data. The study does not take into account the potential heterogeneity of CAR T cells in terms of CAR T expression or T cell immunophenotype (differentiation, exhaustion, fitness...).

See answer #2 to Reviewer #1.

Author response image 1 shows the average tumor radiance on day zero (when CAR-T cell therapy was administered) for all mice. While there is some spread, most mice had tumor localized to the liver or bone marrow.

**Reviewer #3 (Public review):**
Summary:The paper "The 1000+ mouse project: large-scale spatiotemporal parametrization and modeling of preclinical cancer immunotherapies" is focused on developing a novel methodology for automatic processing of bioluminescence imaging data. It provides quantitative and statistically robust insights into preclinical experiments that will contribute to optimizing cell-based therapies. There is an enormous demand for such methods and approaches that enable the spatiotemporal evaluation of cell monitoring in large cohorts of experimental animals.Strengths:The manuscript is generally well written, and the experiments are scientifically sound. The conclusions reflect the soundness of experimental data. This approach seems to be quite innovative and promising to improve the statistical accuracy of BLI data quantification.This methodology can be used as a universal quantification tool for BLI data for in vivo assessment of adoptively transferred cells due to the versatility of the technology.Weaknesses:No weaknesses were identified by this Reviewer.
**Recommendations for the authors:**

**Reviewer #1 (Recommendations for the authors):**
In this paper, the authors propose a significant advancement in optical image data analysis by employing automation. They effectively demonstrate the valuable insights that can be gained from analyzing extensive datasets with a more unbiased methodology. At present, I do not have any specific suggestions for improvement.However, it is important to note that this work is limited in its operational scope. Specifically, it relies on predefined ROIs rather than aligning the signal site with anatomical systems. The scaling model and image cropping are simplistic, animal pose is not taken into account, and the data output needs to be called semi-quantitative or qualitative, and would have been stronger utilizing an AI agent. Nevertheless, this work underscores the potential of automated systems in preclinical image analysis, which is a crucial step towards developing more sophisticated approaches to optical image data analysis.

While our analysis used predefined ROIs, the maRQup pipeline allows users to manually draw ROIs on the mouse image.

**Reviewer #2 (Recommendations for the authors):**
The writing and presentation of data are clear and accurate, but some additional information should be added regarding the imaging protocol used to acquire the original data.The authors mention fluorescence in Figure 1. I expected all the data to be generated from bioluminescent NALM-6 tumors, since bioluminescence is indeed measured in average radiance and can be per pixel (p/sec/cm2/sr/pixel). Fluorescence should be measured using radiance efficiency (p/sec/cm2/sr)/(µW/cm2), a unit that compensates for non-uniform excitation light pattern in the instrument. Would the author find different results if fluorescence data were analyzed separately?

Reviewer #2 is correct that the unit for fluorescence would be radiance efficiency. The word “fluorescent” was included in the label of Figure 1a to highlight that our workflow could be applied to other types of light-generating methods (i.e., fluorescence vs. bioluminescence). However, in this study, measurements of bioluminescent tumors only were analyzed. If fluorescence measurements are to be analyzed, our methods of image acquisition and processing would be directly applicable.

Did the author ever check the signal of the snout in mice with no tumor?

In mice with no tumor, there is no detectable signal in the snout (or anywhere else, for that matter).

The urine of mice contains phosphor, and might give a background signal, especially if longer exposure is used at the end of the study.

For the mice with no tumor injection, the luminescence signal was below background (<10^2^ p/sec/cm^2^/sr/pixel). In particular, we do not detect any signal in the bladder/urine. Additionally, as described in the Supplemental Methods and Figure 1b, only pixels that were on the mouse as determined from the brightfield image were used to calculate the tumor burden from the radiance of the luminescent image. This method ensures that any background signal (e.g., from phosphor in mouse urine) would be excluded in the radiance quantification and not bias the results.

Additionally, as described in the Methods, the exposure time was held constant at 30 seconds for each IVIS measurement across all 37 experiments.

The data using more than 2 million cells comes from only 10 mice, and maybe the biological relevance of this group is limited since it will not be achievable and translatable in humans (PMID: 33653113).

We appreciate Reviewer #2’s attention to this issue. The effect observed in our study is large enough to reach statistical significance despite the small number of mice. Note that the dosing regimen used was optimized for the murine NSG model and would require appropriate scaling before clinical application. Nonetheless, NSG mice remain the gold standard for pre‑clinical in vivo evaluation and their use is generally required by regulatory agencies, such as the FDA, for assessing novel CAR‑T cell therapies; thus these findings are relevant for advancing such treatments.